# Genomic Analysis of Gastrointestinal Parasite Resistance in Akkaraman Sheep

**DOI:** 10.3390/genes13122177

**Published:** 2022-11-22

**Authors:** Yunus Arzik, Mehmet Kizilaslan, Stephen N. White, Lindsay M. W. Piel, Mehmet Ulaş Çınar

**Affiliations:** 1Department of Animal Science, Faculty of Agriculture, Erciyes University, Kayseri 38039, Turkey; 2International Center for Livestock Research and Training, Ministry of Agriculture and Forestry, Ankara 06852, Turkey; 3Department of Veterinary Microbiology & Pathology, College of Veterinary Medicine, Washington State University, Pullman, WA 99164, USA; 4USDA-ARS Animal Disease Research, 3003 Animal Disease Biotech Facility, Washington State University, Pullman, WA 99164, USA

**Keywords:** parasite resistance, heritability, QTL, GWAS, ovine

## Abstract

Genome-wide association studies (GWAS) have been used as an effective tool to understand the genetics of complex traits such as gastrointestinal parasite (GIP) resistance. The aim of this study was to understand the genetics of gastrointestinal parasite (nematodes, *Moniezia spp.*, *Eimeria spp.*) resistance in Akkaraman sheep by performing genomic heritability estimations and conducting GWAS to uncover responsible genomic regions. This is one of the first studies to examine the genetic resistance of Akkaraman sheep to the tapeworm parasite. The samples from 475 animals were genotyped using the Axiom 50K Ovine Genotyping Array. Genomic heritability estimates ranged from 0.00 to 0.34 for parasite resistance traits. This indicates that measured phenotypes have low to moderate heritability estimates. A total of two genome-wide significant SNP associated with *TNEM3* and *ATRNL1* genes and 10 chromosome-wide significant SNPs related with 10 genes namely *NELL1*, *ST6GALNAC3*, *HIPK1*, *SYT1*, *ALK*, *ZNF596*, *TMCO5A*, *PTH2R*, *LARGE1*, and *SCG2* were suggested as candidates for parasite resistance traits. The majority of these candidate genes were involved in several basic biological processes that are essential and important for immune system functions and cellular growth; specifically, inflammatory responses, cellular transport, cell apoptosis, cell differentiation, histone de-acetylation, and endocytosis. These results have implications for animal breeding program studies due to the effect that the genetic background has on parasite resistance, which underlies many productive, health, and wellness-related traits.

## 1. Introduction

Gastrointestinal parasite (GIP) infections are among one the most important health problems in pasture-based livestock production systems [1]. Tapeworm (*Moniezia spp*.), coccidian (*Eimeria spp*.), and nematode (*Trichostrongylus spp*., *Teladorsagia circumcincta*, *Haemonchus contortus*, *Cooperia spp*., *Oesophagostomum spp*., *Chabertia ovina, Bunostomum trigonocephalum,* and *Nematodirus spp*.) parasites are the most common parasites detected in sheep in Turkey, similar to many other countries. GIP infections adversely affect productivity and quality of welfare in animals through clinical signs like weight loss, anemia, and lethargy. Animals at an early growth stage cause considerable damage to both the farm economy and the national economy when carrying GIP, due to effects such as growth retardation.

The economic damage caused by parasitic infection is twofold: directly, is the loss of animal productivity, such as a diminished growth phase, weight loss, and deaths in juvenile lambs; indirectly, is the cost of intervention strategies which consist of chemical drug and veterinary service costs. It has been reported that the annual economic loss due to parasitic diseases in Australia, where sheep and goat farming is intensive, is around $436 million [2]. Moreover, a 10% decrease in the live weight gain of lambs within the growth period is suggested to cost £4.40 per lamb [3]. Considering this situation, economic losses due to parasitic infections should not be ignored during sustainable production within the sheep industry. In recent years, animal breeding has been aimed at obtaining more resistant herds by including complex disease resistance traits in selection indexes, where this is the disease control strategy of many countries, besides other sustainable control strategies such as biological control [4,5,6]. 

It is imperative to understand the genetic parameters (i.e., genetic variance, heritability, and genetic covariance) of measured traits in order to achieve rapid and effective genetic progress for population production (i.e., milk and meat yield, etc.) and wellness (i.e., heat stress, disease resistance, etc.) traits. Moreover, understanding these genetic parameters will allow for more accurate estimate breeding values (EBV) to be calculated and, accordingly, will generate more accurate selection decisions for a population [7,8,9]. Studies on genetic parameter estimation of gastrointestinal parasite resistance phenotypes have mostly focused on nematode species, due to their high economic impact. This has left studies on coccidian and tapeworm infections rather limited. Studies conducted on different nematode species within different sheep breeds have shown that gastrointestinal nematode resistance has a low to moderate varying heritability [10,11,12,13,14,15]. More specifically, a study conducted on Scottish Blackface sheep determined heritability estimates of fecal *Strongyles spp*., *Nematodirus spp*. eggs and Coccidian (*Eimeria spp.)* oocysts were found to be 0.17, 0.09, and 0.09, respectively [16]. Additionally, a study conducted in Texel, Ile France, Morada Nova sheep and crossbreds of this breed, reported that tapeworm (*Moniezia expansa*) egg and coccidian (*Eimeria spp.*) oocyst numbers varied between genetically different animals [17]. Importantly, within each of these studies, the reported heritability of phenotypes associated with GIP resistance has maintained a range from low to moderate. 

Recent technological developments have enabled the effective use of molecular genetics in animal husbandry [18,19,20,21,22,23]. More detailed mapping of the sheep reference genome has led to the development of the Genome-Wide Association Analysis (GWAS) method which associates recorded phenotypes with high-density polymorphisms (Single Nucleotide Polymorphism-SNP) that span the entire genome [24]. This method enables the identification of genomic regions which are responsible for complex traits, such as production and health-related traits, that have low heritability and are therefore influenced by the small effects of many genes (multi-genic) [9,21,25]. According to recent research, sheep and goats have individual resistance and resilience to parasitic infections [26,27]. Many genomic regions, SNPs, and Quantitative Trait Loci (QTL) associated with gastrointestinal parasite resistance-related traits have been reported across studies in different sheep breeds using SNP panels obtained by microarray technologies [28,29,30,31,32]. Genome-wide association studies in sheep have reported that genes associated with acquired immune response, mucus secretion, and hemostasis are directly associated with parasite resistance [30]. For instance, an Australian study deciphering sheep variance in fecal egg counts employing the GWAS methodology with 600,000 SNPs determined that the QTL explaining the high variance was located within chromosome 2 [18]. Furthermore, in a study conducted on Turkish native breeds, namely Karacabey Merino, Kıvırcık, İmroz (Gökçeada), Chios, Cine Çapari, and Karakaçan sheep, a total of 14 variants within the *TLR4* gene were found associated with coccidian oocyst count [33]. Search for prior studies associating Tapeworm (*Moniezia spp.*) carriage to sheep genomic loci did not return any considerable results. Therefore, this is one of the first studies to examine the genetic resistance of domestic sheep to the tapeworm parasite. An association analysis employing an SNP panel that represents the entire sheep genome will begin to elucidate the genetic background of GIP resistance. Therefore, the aim of the present study was to investigate the genetic basis of GIP resistance in Akkaraman sheep. For this purpose, genetic parameter estimates and genome-wide association analyses were performed using phenotypes obtained from fecal egg counts and genotypes characterized by a 50K ovine SNP panel. 

## 2. Materials and Methods

### 2.1. Animals, Sample Collection, and Phenotypic Measurements

The study was conducted in Ankara, Turkey (39°41′ N, 33°01′ E). The place is known for its continental climate with severe, snowy winters, and dry summers. The average annual precipitation is 389 mm, and the average annual temperature is 11.7 °C. The average altitude of the region is 938 m with most of the pastureland being of low quality. A total of 475 Akkaraman lambs (185 males and 290 females) were used for the study. The animals were randomly selected from three herds that are members of the National Community-based Small Ruminant Breeding Program. Mating strategies are applied within these herds based on pre-weaning growth traits. Lambs were born in January and February 2021 and weaned at approximately three months of age. After weaning, 370 lambs were kept on pasture during the summer without supplemental feeding, while 105 lambs received 750–1000 g concentrate feed per day during the 90-day finishing period. 

Environmental factors (i.e., sex, herd, birth type, feeding type, location) were recorded for animals recruited into the study. Fecal samples were collected from 129 animals at three months of age (May 2021) and 478 animals at six months of age (August 2021), including the earlier 129 subsets. A total of 20–30 grams of feces were collected directly from the rectum of the animals to minimize contamination and it was ensured that at least 60 days had passed since the last anti-parasite treatment of animals. Samples were transported to the laboratory in a cold chain and stored at 4 °C until the feces were examined. During the August sampling, approximately 6 ml of blood (in vacuum EDTA tubes) was collected from the jugular vein of the animals.

GIP resistance phenotypes in sheep were based on fecal egg counts (FEC) of three different parasite species. The phenotypes for nematode egg counts at three months and six months of age were **NemFEC3** and **NemFEC6**, respectively. The phenotypes for *Moniezia spp*. (tapeworm) at three months and six months of age were **MonFEC3** and **MonFEC6**, respectively. Finally, the phenotypes for the number of fecal oocysts of coccidian parasites were **CocFOC3** at three months of age and **CocFOC6** at six months of age. The number of eggs of nematodes, tapeworms, and oocysts of coccidian parasites was counted using the McMaster technique [34]. Nematode egg counts were enumerated as a binary trait (i.e., 0 = control, 1 = case) and the other traits were calculated as continuous traits. The quantitative responses were checked for outliers and the exceeding observations (the mean ± 3*standard deviations) were removed from the data.

### 2.2. DNA Extraction, Genotyping, and Quality Control (QC) 

The blood samples were transferred to International Center for Livestock Research and Training (ICLRT), Genetic Laboratory, for DNA extraction. To avoid contamination risk, DNA extraction was implemented with the Qiacube HT automated device using a commercial kit (Qiagen Blood kit, Hilden, Germany). Quality control of the extracted DNA was performed before further analysis, where samples meeting the quality criteria (A260/280 > 1.8, A260/230 >1.5, >70 ng/µL) were kept at −20 °C until genotyping. Genotyping was performed using the Axiom 50K Ovine Genotyping Array (Thermo Fisher Sci, Waltham, MA, USA) with the GeneTitane MultiChanel instrument at the ICLRT Genetics Laboratory. The 96-array format of the Axiom 2.0 Assay was followed as per manual workflow protocols. 

Genotyping quality control (QC) began with the Axiom Analysis Suite program. In addition to the Axiom Analysis Suit program, one more quality control step was applied to the data by using the “check.marker” function of the GenABEL package within R [35,36]. The applied criteria are accepted by many researchers and have been employed in a multitude of studies [37,38,39]. Criteria included removal of SNPs with a call rate lower than 95%, a minor allele frequency (MAF) lower than 0.05 (5%), association with the X chromosome, and deviation from HWE (Hardy-Weinberg Equilibrium) at *p* < 0.05. Furthermore, samples that had a lower call rate than 90% and identity by state (IBS) higher than 95% were removed from the data set. 

### 2.3. Statistical Analysis

Imputation was applied to the quality-controlled genomic data before proceeding to genomic heritability estimation and genome-wide association analyses. SNPs that did not meet QC following genotyping were imputed according to the current status of this SNP in the population. The expected maximization (EM) algorithm was used in this study and the process was carried out by R GenABEL packages [35].

The Animal Mixed Model and ‘Restricted Maximum Likelihood’ (REML) approach were used to estimate genomic heritability by using the ‘sommer’ package of R [40]. Measured environmental factors were included in the model as a fixed effect and the genomic relationship matrix was included in the model to capture the covariance between individuals in terms of the phenotypes of interest. The animal mixed model is given below:y=µ+Xβ+Zu+e
$$V=\left[\begin{array}{ccc} Z_{\;}G\sigma_{u_{\;}}^{2}Z^{\prime }+I\sigma_{e_{\;}}^{2} & \cdots & Z_{\;}G\sigma_{u_{\;}}Z^{\prime }+I\sigma_{e_{\;}}\\ \vdots & \ddots & \vdots \\ Z_{\;}G\sigma_{\;}Z^{\prime }+I\sigma_{e_{\;}} & \cdots & Z_{\;}G\sigma_{u_{\;}}^{2}Z^{\prime }+I\sigma_{e_{\;}}^{2} \end{array}\right]$$
where y is the vector of individual observations of each trait focused, **µ** is the population mean regarding the trait of interest, **β** is the vector of fixed environmental effects such as sex (2 levels), birth type (2 levels), birth month (3 levels), u is the polygenic background effect obtained from MVN (*u* ~ 0, $G\sigma_{u}^{2}$) and e is the vector of random residual errors obtained from MVN (*e* ~ 0, $I\sigma_{e}^{2}$). Meanwhile, **X** and **Z** are the design matrices mapping fixed effects and polygenic background effects to each observation, respectively. Furthermore, $\sigma_{u}^{2}$ and $\sigma_{e}^{2}$ are the additive genetic variance and random residual variances for the particular trait, respectively. Lastly, **G** represents the genomic relationship matrix derived by VanRaden (2008) [41]. The formula for the G matrix is as follows:$$G=\frac{ZZ^{\prime }}{2\sum^{\;}p_{i}\left(1-p_{i}\right)}$$

After estimating the variance component of each trait, the narrow sense heritability estimates of each trait were obtained by using the model given below;
$$h^{2}=\frac{\sigma_{u}^{2}}{\sigma_{p}^{2}\left(\sigma_{u}^{2}+\sigma_{e}^{2}\right)}$$
where $h^{2}$ is the heritability estimate of each trait, $\sigma_{u}^{2}$ is the estimate of genetic variance for each trait, $\sigma_{p}^{2}$ is total phenotypic variance and $\sigma_{e}^{2}$ is the error variance. The environmental factors described above are included in the model as a fixed effect.

The same general model was used to estimate the SNP effect in genome-wide association studies. In the model, the additive effect estimated with the genomic relationship matrix (SNP-based identity-by-state) was included to account for the fixed genotypic effect of each SNP. A widely used graphical method, the Manhattan plot, was used to visualize the *p*-values showing the significance of the SNP effects. The *p*-values of the SNP effects were calculated based on the association analyses by ordering the position information of the SNPs on each chromosome on a −log10 scale. A statistical significance threshold value was used to avoid Type 1 Error (probability of rejection if the null hypothesis is true) accumulation in multiple SNP tests. The approach of Bonferroni correction for multiple tests was used to determine genome- and chromosome-wide significance thresholds for the traits.

### 2.4. Functional Annotation of Significant SNPs

Using the Oar_v4.0 genome assembly, we managed to get positional information for associated SNPs and candidate genes from the NCBI Genome Data Viewer [42]. SNPs that overlapped genes designated those genes as candidates. Furthermore, if they found SNP was not located within a gene, then the nearest gene was suggested as a candidate. DAVID Bioinformatics Resources 2021 was used to obtain biological information on the identified candidate genes [43]. Additionally, previously identified QTLs related to sheep parasite resistance were examined through the Animal QTL Database to see if there was any overlap with the SNPs identified in this study [44]. In case of insufficient annotation in the sheep reference genome, orthology between species was used, where the annotation of specific genes from cattle, goats, mice, and humans was evaluated for shared biological function. Finally, metabolic pathways involving candidate genes were regarded by Gene Ontology terms, which were explored on the QuickGo website [45].

## 3. Results

### 3.1. Descriptive Statistics

Fecal egg counts for nematodes and tapeworms and fecal oocyst counts for coccidian parasites were quantified per gram of feces for 129 animals at three months of age and 475 animals at six months of age. Descriptive statistics and prevalence are shown in Table 1 and Appendix A.

### 3.2. SNP Marker Summary

The number of samples for each tested trait is reported in the descriptive statistics table (see Table 1). The raw genotype data contained 49,931 SNPs, which was reduced to 40,463 SNPs after applying the QC criteria mentioned in the Material and Methods section. During genotyping, only one animal failed the QC analyses within the Axiom Suit Analyzer, due to its poor-quality scores. All samples passed the QC analyses performed using the GenABEL package, but approximately 9,000 SNPs failed to meet the SNP criteria. 

### 3.3. Genomic Heritability Estimates 

Heritability estimates for parasite resistance phenotypes ranged from low to moderate in Akkaraman sheep. As can be seen in Table 1, the lowest heritability was calculated for MonFEC3 (0.00) and NemFEC6 (0.009). Heritability estimates for the traits NemFEC3, CocFOC3, MonFEC6, and CocFOC6 were found to be moderate at 0.34, 0.11, 0.30, and 0.25, respectively. When the margins of error of the heritability estimates were examined, it was found that the errors of the heritability estimates for the traits collected at three months of age were higher than the errors for the traits collected at six months of age (see Table 1).

### 3.4. Genome-Wide Association Studies 

Genome-wide association analysis was performed for six different traits (namely NemFEC3, MonFEC3, CocFOC3, NemFEC6, MonFEC6, and CocFOC6) belonging to three different parasite groups (i.e., nematodes, tapeworm and coccidian parasite) associated with GIP carriage. A Linear Mixed Model (LMM) was used to identify important SNPs associated with phenotypes. The greatest advantage of LMMs in GWAS analysis is that false positive results can be eliminated by employing population stratification and animal relationships. The genotypic effect of SNPs and the random genetic effect (additive effect) of the animal were employed to calculate the SNP-based genomic relationship matrix (G). The dependent variable in the LMM equation was the measured phenotypes. SNP effects and environmental factors were included in the model as fixed effects, whereas the individual effects of animals and the effect arising from the genetic background of the animal were included in the model as random effects.

The observed test statistics for each SNP were compared to the test statistics expected under the null hypothesis using quantile-quantile (Q-Q) plots (Figure 1). A Q-Q plot and the estimated inflation factor lambda (λ) were attained for each phenotype which was set to 1 with genomic control applied to *p*-values as defined by Devlin and Roeder, 1999. Manhattan plots were used to visualize the −log10 (*p*-values) of all SNPs respective to their position on the relevant chromosome (Figure 2). Genome- and chromosome-wide significance thresholds defined by using Bonferroni correction were implemented to avoid increasing the Type 1 Error rate due to testing of multiple SNPs [46]. Accordingly, the genome-wide significant threshold was 1.23 × 10^−6^ (0.05/40,463) and the chromosome-wide significant threshold was 3.21 × 10^−5^ (1/40,463).

The age at which the lambs were sampled was determined to have a significant impact on all traits associated with parasite resistance. The one exception to this was for the MonFEC6 trait. The herd effect was found to be significant for the MonFEC3 and NemFEC3 traits, while the sex of the animal was found to only affect the MonFEC3 trait. Notably, no environmental factor was found to affect the MonFEC6 trait. Finally, for the CocFOC3 trait, there was an interaction effect between the feed type and sex, while the NemFEC6 trait had a three-way interaction between the herd, sex, and feed type.

A total of 12 SNPs were found to be statistically significant as a result of the GWAS studying traits associated with GIP resistance (see Table 2). Two of these SNPs on sheep chromosomes (OAR) 22 and 26 were found to be genome-wide significant for NemFEC6 and MonFEC6 traits. The remaining 10 SNPs were found to be significant at a chromosome-wide level. The 12 SNPs found to be statistically significant were located on six different sheep chromosomes (OAR1, 2, 3, 7, 21, 22, and 26). On the other hand, all these SNPs are located on or near (±200 kb) the genes defined within the sheep genome.

### 3.5. Candidate Genes and QTLs

Using the Oar_v4.0 genome assembly, we obtained positional information for associated SNPs and candidate genes from the NCBI Genome Data Viewer [42]. Candidate genes were defined as those which overlapped or were near significant SNPs. As a result of the GWAS studies for the traits associated with GIP resistance, a total of 12 different SNPs were detected on six different chromosomes. Seven of these SNPs were located within an annotated gene namely Neural Epidermal Growth Factor-Like 1 (*NELL1*), Teneurin Transmembrane Protein 3 (*TENM3*), ST6 N-Acetylgalactosaminide Alpha-2,6-Sialyltransferase 3 (*ST6GALNAC3*), Attractin Like 1 (*ATRNL1*), Homeodomain Interacting Protein Kinase 1 (*HIPK1*), Synaptotagmin 1 (*SYT1*) and ALK Receptor Tyrosine Kinase (*ALK*) in the genome. The other five of these SNPs are located in regions nearby specific genes namely Zinc Finger Protein 596 (*ZNF596*), Transmembrane and Coiled-Coil Domains 5A (*TMCO5A*), Parathyroid Hormone 2 Receptor (*PTH2R*), LARGE Xylosyl- and Glucuronyltransferase 1 (*LARGE1*), and Secretogranin II (*SCG2*) on the genome.

## 4. Discussion

Pasture-based production systems which raise small ruminants have long incurred economic losses due to parasitic diseases. Global warming and associated climate change have increased the frequency of parasite infections and the diversity of pathogens in a region. On the other hand, the resistance of parasites to anti-parasitic drugs is increasing day by day. Considering all these factors together, it will be more difficult to reduce parasite infections in farm animals in the future, and alternative strategies for control will be needed. While there are present studies focused on the production and reproductive traits of the native Akkaraman sheep breed and its crossbreeds, there are not enough studies on the genetics of health traits such as parasite resistance [9,47,48,49,50,51].

### 4.1. Genomic Heritability Estimates

In our study, we estimated the heritability of egg/oocyst counts of nematodes, tapeworms, and coccidian parasites in Akkaraman lambs at three and six months of age. Heritability estimates for the traits ranged from 0 (MonFEC3) to 0.34 (NemFEC6). Overall, it was found that the error margin of the heritability estimates for traits measured at three months of age was higher than that of the traits at six months of age. One possible reason for this could be that the animals were not sufficiently exposed to parasites during this period. In relation to this, it is possible that animals that are three months old may not have had enough time to acquire a sufficient or remarkable egg/oocyst load, where it is known that the immune system is not fully developed at this time. Thus, it can be assumed that animals of this age do not adequately reflect the effects of genetic potential on their immune system. Lastly, it is feasible that the small sample size affected the statistical power of the analysis.

Studies estimating genetic parameters of indicator phenotypes for resistance to gastrointestinal parasites have mostly focused on the nematode species because of their high economic damage. Prior studies conducted on varying nematode species in alternate sheep breeds have calculated heritability estimates ranging from low to moderate when assessing resistance to gastrointestinal nematodes [10,11,12,13,15,52]. This is mirrored within the current study, where the heritability for the total number of nematode eggs in three- and six-month-old lambs was reported as 0.34 and 0.01. A study on *Haemonchus contortus* FEC in Santa Ines sheep, raised in a tropical climate, provided a heritability estimate of approximately 0.10 [10]. Another study reported a heritability range from 0.01 to 0.22 in Scottish Blackface lambs aged from one to six months old when calculating the fecal egg count heritability for nematodes [11]. In addition, the heritability of FEC phenotypes in the pre- and post-drench period (anti-parasitic treatment) in Avikalin and Malpura ewes was found to be 0.21–0.03 and 0.19–0.12, respectively [12]. Further support of the calculated heritability comes from a study conducted on Spanish Churra dairy ewes, where the heritability of the fecal egg count trait for *Teladorsagia circumcincta* ranged from 0.09 to 0.12 [13] (Gutiérrez-Gil et al. 2010). Lastly, studies in the Merino sheep breed, which is the most common breed in the world, suggested that the heritability for the FEC trait was between 0.18 and 0.10 [15,52].

Studies on coccidian and tapeworm infections are rather sparse compared to nematode studies. In a study conducted in Texel, Ile France, Morada Nova sheep, and their crossbreds were reported to carry alternate numbers of tapeworm (*Moniezia expansa*) eggs and coccidian (*Eimeria spp.*) oocysts [17]. Finally, in a study of Scottish Blackface sheep, the heritability estimates of fecal *Strongyles spp.* eggs, *Nematodirus spp.* eggs and Coccidian (*Eimeria spp.*) oocysts were found to be 0.17, 0.09, and 0.09, respectively [16]. Considering the age groups used in the study on Scottish Blackface sheep, the heritability estimation for coccidian in three-month-old lambs was found to be similar to the heritability (0.11) obtained in our study. It is expected that animals residing in heavily contaminated areas have obtained a gene-environment relationship allowing for alternate genetic resistance to diseases.

### 4.2. Genome Wide Association Studies (GWAS)

As a result of the GWAS, 12 distinct SNPs were found to be statistically significant for measured traits associated with parasite resistance. While two of these SNPs were significant at the genome-wide threshold, 10 SNPs were found to be significant at the chromosome-wide level. The most significant SNP (*rs421027634*) was found on OAR26 for the NemFEC6 trait with a *p*-value of 1.08 × 10^−7^. The SNP (*rs401554073*) with the second lowest *p*-value (6.92 × 10^−7^) was associated with the MonFEC6 trait and was found on OAR22. Considering that the phenotypes belong to three different parasite groups acquired at three and six months of age, no SNPs with pleiotropic effects associated with these traits were found. This situation may be a result of the immune system acquiring alternate defense mechanisms against alternate parasite species. Significant SNPs were annotated according to their positions on the reference genome (OAR_v4.0) using the National Center for Biotechnology Information (NCBI) Genome Data Viewer. While seven of the SNPs were located within a particulate gene defined within the genome, five of them were found nearby at least one particular gene. Accordingly, the suggested candidate genes for parasite resistance were *NELL1*, *TENM3*, *ST6GALNAC3*, *ATRNL1*, *HIPK1*, *SYT1*, *ALK*, *ZNF596*, *TMCO5A*, *PTH2R*, *LARGE1*, *SCG2*.

For the NemFEC6 trait, *TENM3* was found to be associated with SNP rs421027634 and had the lowest *p*-value (in the scale of −log10) as a result of GWAS. The resultant protein from this gene is a teneurin protein, which is employed in multiple cell components, for example, the cell membrane (GO:0016020), and takes part in many biological functions, such as cell-cell adhesion (GO:0098609), cell adhesion (GO:0007155) and signal transduction (GO:0007165). Studies show that the Wnt signaling system, which moderates the innate immune response and adaptive immune response with the production of inflammatory cytokines, upregulates the Teneurin-3 protein [53]. This family of conserved proteins may also play a role in both wound-repairing and signal transduction following cell damage in the alimentary tract by larval and adult parasites in nematode infections. *TENM3* has been identified in viral control (Holmes et al 2020). Further, *TENM3* has been associated with hip dysplasia, suggesting a role in development (Xu et al 2021). *ST6GALNAC3* was another associated gene for the NemFEC6 trait located on OAR1. This gene is a member of sialyltransferase family, which transfers sialic acids from CMP-sialic acid to terminal carbohydrate groups on glycoproteins, supporting the active part *ST6GALNAC3* plays in the glycoprotein metabolic process (GO:0009100) [54]. Glycoproteins (i.e., CD8, CD4) are released from cytotoxic T (Tc) cells of the adaptive immune system to aid in killing viruses or other pathogen-infected cells. Other immune-based glycoproteins serve to provide essential signals that activate the maturation of B cells, Tc cells, and macrophages [55]. The last gene found associated with the NemFEC6 trait was *TMCO5A* (Transmembrane and Coiled-Coil Domains 5A). This gene encodes a protein particularly integral to membrane components (GO:0016021) and plays a role in protein binding (GO:0005515). 

After the GWAS analysis, *PTH2R* and *ATRNL1* genes were found to be associated with the fecal egg count phenotypes of *Moniezia spp.* (MonFEC3 and MonFEC6) on OAR2 and 22, respectively. The *rs401554073* SNP was associated with the *ATRNL1* gene at a genome-wide significance level with a *p*-value of 6.92 × 10^−7^. While only one of the present genes was significant at a genome-wide level, it is interesting to note that both of these genes are directly related to G-Protein-Coupled Receptors (GPCRs). While *PTHR2G* gene is a member of the G-protein coupled receptor 2 family, the *ATRNL1* gene plays an active role upstream of or within G protein-coupled receptor signaling pathways. More specifically, the protein encoded by *PTR2G* is located within the cell membrane (GO:0005886, GO:0005887, GO:0016020, and GO:0016021) and actively involved in cell surface receptor signaling (GO:0007166), G protein-coupled receptor signaling (GO:0007186) and adenylate cyclase-modulating G protein-coupled receptor signaling (GO:0007188). Similar to the *PTHR2G* gene, *ATRNL1* codes for a cell membrane protein involved in G protein-coupled receptor signaling pathways (GO:0007186). By acting as gate-keepers, these GPCRs play an important role in the induction of cytokines and other signals responsible for the stimulation and manipulation of the adaptive immune system [56]. They specifically shape macrophage immune responses to extracellular pathogens as well as injury-related danger molecules [57,58]. Importantly, *ATRNL1* has been identified in viral responses (Wang et al 2021) with a potential impact on immune responses (Zhang et al 2019). GPCRs play an important role in recognizing pathogens and antigens by macrophages, which are critical to innate immune system function. Therefore, it can be thought that the recognition of tapeworm infections and the resultant immune response can be related to the density of these receptors. GPCRs, and therefore the *PTH2R* and *ATRNL1* genes, were suggested as candidate genes for resistance to *Moniezia spp.* infection in sheep.

As a result of GWAS for CocFOC3 and CocFOC6, four important genes namely *HIPK1*, *SYT1*, *SCG2,* and *ALK* were discussed as partaking in sheep resistance to coccidian parasite infection. *ALK* is a protein-coding gene responsible for the regulation of the dopamine receptor signaling pathway (GO:0060159). Dopamine is a neurotransmitter that actively plays a role in the regulation of the immune system [59]. In studies, it has been reported that dopamine levels increase in the brain of chickens during cases of infection or heat stress; however, dopamine levels in the brain were decreased in broiler chickens exposed to *Eimeria spp.* [60]. Conspicuously, assessment of *HIPK1, SCG2,* and *ALK* genes identified during GWAS analysis for Coccidian parasite resistance traits (CocFOC3 and CocFOC6), characterized these genes as partaking in the biological process of cell apoptosis. In particular, the *HIPK1* gene plays a role in the endothelial cell apoptotic process (GO:0072577) and extrinsic apoptotic signaling pathway (GO:0097191), where the *SCG2* gene directly downregulates these same two processes (GO:2000352 and GO:2001237). During coccidiosis, sporozoites are ingested and penetrate the epithelial cells to settle into their replicative niche. These sporozoites divide asexually and form merozoites as a result of this division. After a number of generations, the parasite has increased enough in number to rupture the host cell, resulting in cell death [61]. In a study, it was reported that intestinal parasites activate intrinsic and extrinsic factors in intestinal epithelial cells, causing the cell to undergo apoptosis [62]. It is feasible, that genes related to cellular apoptosis are associated with *Eimeria spp.* resistance due to the effect of merozoites in the cell on apoptotic (intrinsic and extrinsic) factors in the cell. Alternatively, *ALK* and *SCG2* genes were also identified as being associated with the mitogen-activated protein kinase (MAPK) signaling pathway, where both genes are involved in the positive regulation of said cascade (GO:0043410 and GO:0000165, respectively). It has been reported that MAPK signaling is suppressed in host intestinal epithelial cells in an infection caused by *Cryptosporidium*, which is a *protozoan* parasite that infects intestinal epithelial cells similar to the *Eimeria* species [63]. Along with apoptotic and MAPK pathways, *SCG2* has been determined to play a very active role in the formation of innate immune and adaptive immune responses in the body. In particular, *SCG2* is involved in biological processes such as inflammatory responses (GO:0006954), eosinophil chemotaxis (GO:0048245), positive chemotaxis (GO:0050918), induction of positive chemotaxis (GO:0050930) and endothelial cell migration (GO:0043542). Within human studies, the *HIPK1* gene is associated with chronic inflammatory diseases such as Crohn’s disease, primary sclerosing cholangitis, and ulcerative colitis, all of which localize to the gastrointestinal tract [64]. 

In general, many genes identified within the present study are heavily intercalated within basic biological functions that are essential to inflammation. These include processes such as eosinophil chemotaxis, inflammatory response, cell apoptosis, immune-related protein synthesis, and intercellular signaling. It was also noted that candidate genes responsible for similar host cell processes were grouped within parasite species. For instance, the *PTH2R* and *ATRNL1* genes found to be related to tapeworm numbers were involved in the synthesis of G-Protein-Coupled Receptors (GPCRs). Furthermore, *HIPK1*, *SCG2,* and *ALK* genes were all associated with cellular apoptosis, which may occur as a result of coccidian infection of the intestinal epithelial cell. Alternatively, *SCG2* and *ALK* genes associated with coccidian parasite resistance are also actively involved in the MAPK signaling pathway, which has a significant role in immune responses against parasite infection.

## 5. Conclusions

In conclusion, genomic heritability for parasite resistance traits was estimated and found to be low to moderate. Furthermore, GWAS returned two genome-wide and 10 chromosome-wide significant SNPs for measured parasite resistance traits. In this study, we provided possible explanations and meaningful results about the genetics of parasite resistance. Considering that parasite resistance is directly related to the health and survivability of sheep, the resultant data may promote the development of breeding approaches that may produce more resistant and sustainable animals against disease, leading to remarkable economic impact. Similar studies with larger animal numbers and encompassing different sheep populations are required in order to transfer the present knowledge to employed applications against present parasites. The next step would be a sequencing of candidate genes in order to identify the SNPs responsible for this resistance in order to use them more easily in a marker-assisted selection approach.

## Figures and Tables

**Figure 1 genes-13-02177-f001:**
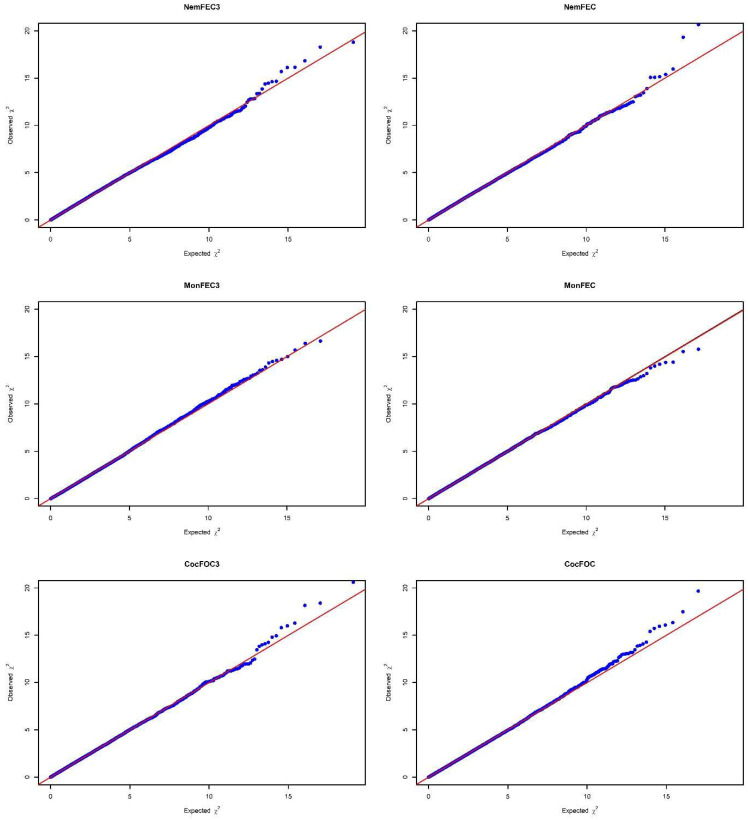
Quantile-quantile (Q-Q) plots of genome-wide association studies (GWAS) for the parasite resistance traits.

**Figure 2 genes-13-02177-f002:**
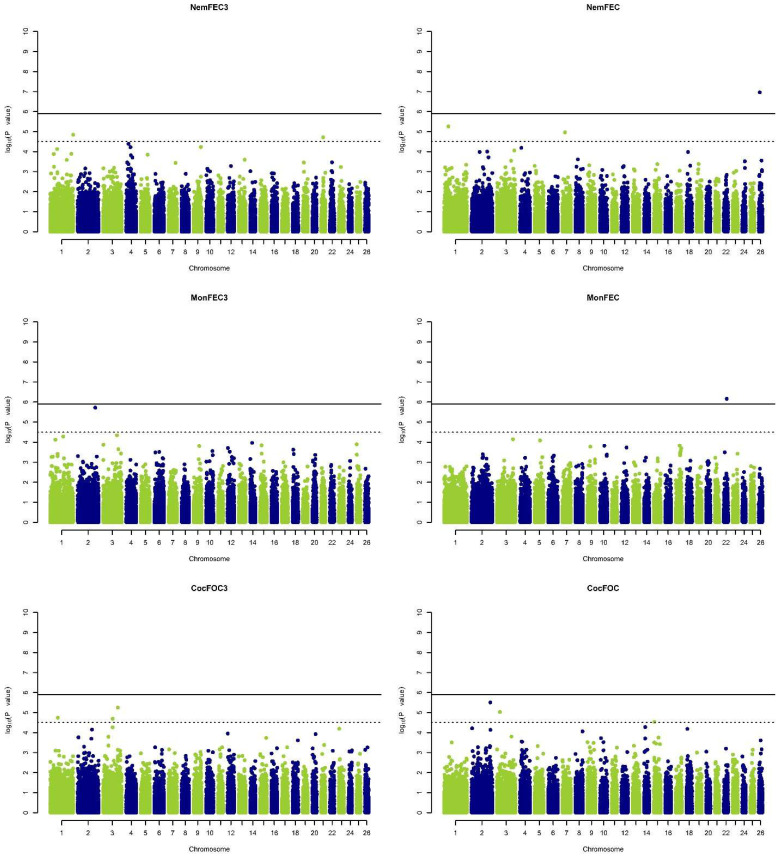
Manhattan plots of six parasite resistance traits. The solid line indicates the genome-wide significance level, and the dashed line indicates the chromosome-wide significance level. Nematode egg count at three months old (NemFEC3), Nematode egg count at six month old (NemFEC6), Tapeworm egg count at three months old (MonFEC3), Tapeworm egg count at six months old (MonFEC6), Eimeria spp. oocyst count at three months old (CocFOC3), Eimeria spp. oocyst count at six months old (CocFOC6).

**Table 1 genes-13-02177-t001:** Descriptive statistics and genomic heritability estimates (*h*^2^) of fecal traits.

Trait	N	Mean	Median	Minimum	Maximum	% 0 Values	SD	*h* ^2^	SE ^1^
**NemFEC3**	129	36.51	0	0	485	68.59	68	0.34	0.28
**MonFEC3**	129	2247	295	0	14,383	38.01	2867	0.00	0.20
**CocFOC3**	129	1790	978	0	38,203	4.95	3620	0.11	0.24
**NemFEC6**	475	6.68	0	0	271	93.90	28	0.01	0.05
**MonFEC6**	475	2810	0	0	95,841	68.63	8250	0.30	0.11
**CocFOC6**	475	2802	1670	0	23,139	5.68	3104	0.25	0.12

^1^ standard error.

**Table 2 genes-13-02177-t002:** Significant SNP associated with parasite resistance phenotypes.

Traits	SNP Name	Chr.	Position (bp) ^1^	*p*-Value	Associated Genes	
					Name	Distance (kbp)
**NemFEC3**	*rs415401096*	1	275,345,749	1.44 × 10^−5^	*ZNF596*	~5 Kb
**NemFEC3**	*rs408499938*	21	23,300,938	1.90 × 10^−5^	*NELL1*	within
**NemFEC6**	*rs421027634*	26	12,363,158	1.08 × 10^−7^	*TENM3*	within
**NemFEC6**	*rs413573397*	1	51,891,269	5.48 × 10^−6^	*ST6GALNAC3*	within
**NemFEC6**	*rs403250421*	7	30,603,665	1.09 × 10^−5^	*TMCO5A*	~35 Kb
**MonFEC3**	*rs409854037*	2	209,775,617	1.01 × 10^−5^	*PTH2R*	~200 Kb
**MonFEC6**	*rs401554073*	22	34,528,500	6.92 × 10^−7^	*ATRNL1*	within
**CocFOC3**	*rs428814111*	3	177,167,791	5.63 × 10^−6^	*LARGE1*	~200 Kb
**CocFOC3**	*rs406580275*	1	90,278,893	1.79 × 10^−5^	*HIPK1*	within
**CocFOC3**	*rs405203400*	3	114,958,514	2.02 × 10^−5^	*SYT1*	within
**CocFOC6**	*rs405331699*	2	224,516,770	3.05 × 10^−6^	*SCG2*	~200 Kb
**CocFOC6**	*rs412952616*	3	36,311,950	9.18 × 10^−6^	*ALK*	within

^1^ SNP position based on OAR_v4.0.

## Data Availability

The data presented in this study are available on a reasonable request from the corresponding author. The data are not publicly available due to the legal restriction of data deposition regarding indigenous breeds.

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
