# Peer review of "Genomic Analysis of Gastrointestinal Parasite Resistance in Akkaraman Sheep"

_genes, 2022, doi:10.3390/genes13122177_

Round 1

Reviewer 1 Report

1. Line 14: nor only health traits!

2. Line 18: “domestic sheep” should be “Akkaraman sheep”.

3. Line 35: “sheep and goat” should be “livestock”.

4. Line 54: It may be good to emphasize the restrictions on biological control of GIP (doi:10.34248/bsengineering.627196).

5. Line 162. Please mention the quality control of the data more detailed.

6. Line 179: “birth month etc.” please write all exactly.

7. Line 229: in Table 1, please also give median value which is more descriptive on count data.

8. Line 278: The reference (Benjamini and Hochberg, 1995) should be numbered according to journal writing rules.

9. Lines 347-348: “Thus, it can be assumed that animals of this age 347 do not adequately reflect the effects of genetic potential on its immune system” this phrase is important, you can add this sentence to abstract also.

Author Response

Reviewer 1

  1. Line 14: nor only health traits!

The sentence was modified. “health “ is removed.

  1. Line 18: “domestic sheep” should be “Akkaraman sheep”.

“domestic sheep” was replaced by “Akkaraman sheep”.

  1. Line 35: “sheep and goat” should be “livestock”.

It was replaced.

  1. Line 54: It may be good to emphasize the restrictions on biological control of GIP (doi:10.34248/bsengineering.627196).

It was added.

  1. Line 162. Please mention the quality control of the data more detailed.

More detail was given in line between 138-140.

  1. Line 179: “birth month etc.” please write all exactly.

They were written more detailed.

  1. Line 229: in Table 1, please also give median value which is more descriptive on count data.

The column for median of observation was added to Table 1.

  1. Line 278: The reference (Benjamini and Hochberg, 1995) should be numbered according to journal writing rules.

It was corrected.

  1. Lines 347-348: “Thus, it can be assumed that animals of this age 347 do not adequately reflect the effects of genetic potential on its immune system” this phrase is important, you can add this sentence to abstract also.

This sentence was included in the Discussion part as it was mostly for the discussion of the results obtained in the 3-month period and the results obtained in the 6-month period. This sentence is a detailed sentence for abstract and we were also able to give superficial information due to the word limit in abstract.

Reviewer 2 Report

Dear Authors

Very interesting work. Please follow my fwee remarks.

Best regards

Author Response

Reviewer 2

Key words:

The order of key words was corrected according to the suggestion of Reviewer 2.

Table 1:

According to the suggestion of Reviewer 2, the numbers of infected and non-infected animal were presented in additional table (Supplementary Table 1) and added as supplementary materials.

Conclusion:

The suggestion of Reviewer 2 was added to conclusion part.
